# The epidemiology of SARS-CoV-2, Influenza, and Respiratory Syncytial Virus circulation among adult patients with acute respiratory illness in Kampala Metropolitan Area in Uganda

James Arinaitwe[1,2], Barnabas Bakamutumaho[3], Levicatus Mugenyi[1,4], Noah Kiwanuka[2], Patricia Alupo[1,5], Aida N. Kawuma[1,6], Karen Ndahura[1,7], Maria Sekimpi[1], Eva Akurut[1,6], Winters Muttamba[1], Moses Joloba[8], Moses Ocan[9], Eddie Wampande[10], Jacqueline Kyosiimire Lugemwa[3,11], Jane Nakibuuka[12], John Lusiba[1,13], Joseph Okia[14], Darius Owachi[15], Pauline Byakika-Kibwika[16,17], Bruce Kirenga[1,16]*

1 Department of Research and Innovation, Makerere University Lung Institute, Kampala Uganda, 2 School of Public Health, Makerere University, Kampala, Uganda, 3 Vaccine Preventable Diseases-EPI, CDC-UVRI Influenza Surveillance Program, Uganda Virus Research Institute, Entebbe, Uganda, 4 MRC/UVRI & LSHTM Uganda Research Unit, Entebbe, Uganda, 5 University of Groningen, Groningen, Netherlands, 6 Infectious Disease Institute, Makerere University, Kampala, Uganda, 7 ISBAT University, Kampala, Uganda, 8 School of Biomedical Sciences, Makerere University College of Health Sciences, Kampala, Uganda, 9 Department of Pharmacology, Makerere University College of Health Sciences, Kampala, Uganda, 10 College of Veterinary Medicine, Makerere University, Kampala, Uganda, 11 University of Kisubi, Entebbe, Uganda, 12 Mulago National Referral Hospital, Ministry of Health, Kampala, Uganda, 13 Uganda Peoples' Defense Forces, Kampala, Uganda, 14 Northpark Specialist Hospital, Kampala, Uganda, 15 Kiruddu National Referral Hospital, Ministry of Health, Kampala, Uganda, 16 School of Medicine, Makerere University College of Health Sciences, Kampala, Uganda, 17 Mbarara University of Science and Technology, Mbarara, Uganda

* brucekirenga@yahoo.co.uk

## Abstract

### Background

Acute respiratory infections (ARI) caused by viruses such as SARS-CoV-2, influenza, and respiratory syncytial virus (RSV) posed significant public health challenges, particularly in low-income countries. Understanding the co-circulation dynamics of these pathogens is crucial for effective public health monitoring, surveillance, and control interventions. This study aimed to characterize the epidemiology of SARS-CoV-2, influenza, and RSV co-circulation among patients with ARI in Uganda's central business districts.

### Methods

A retrospective cohort study was conducted among 1,265 adult outpatients aged 18 years and older presenting with ARI in Kampala, Wakiso, and Mukono districts. Data were collected for the period from March 2023 through April 2024. Nasopharyngeal swabs were collected and tested for SARS-CoV-2, influenza A/B, and RSV using

**Data availability statement:** All relevant data for this study are publicly available from the figshare repository (https://doi.org/10.6084/m9.figshare.31645411).

**Funding:** The study was funded by the Government of Uganda through the secretariat of Science, Innovation and Technology - office of the president. The funders had no role in study design, data collection and analysis, decision to publish, or preparation of the manuscript.

**Competing interests:** There are no competing interests to declare by any of the authors.

**Abbreviations:** ARI, Acute Respiratory Infection; RSV, Respiratory Syncytial Virus; RT-PCR, Reverse Transcription Polymerase Chain Reaction; LRTI, Lower Respiratory Tract Infection; LMICs, Low- and Middle-Income Countries; SARI, Severe Acute Respiratory Illness; VTM, Viral Transport Medium; NIC, National Influenza Centre; UVRI, Uganda Virus Research Institute; CT, Cycle Threshold; RNA, Ribonucleic Acid; CDC, Centers for Disease Control and Prevention; NP, Nasopharyngeal; OP, Oropharyngeal; RNAseP, Ribonuclease P; MREC, Mulago Research and Ethics Committee; UNCST, Uganda National Council of Science and Technology; SPO2, Peripheral Capillary Oxygen Saturation; HIV/AIDS, Human Immunodeficiency Virus/Acquired Immunodeficiency Syndrome; WHO, World Health Organization; UBOS, Uganda Bureau of Statistics; COVID-19, Coronavirus Disease 2019; SARS-CoV-2, Severe Acute Respiratory Syndrome Coronavirus 2.

RT-PCR. Data on demographics, clinical presentation, and SARS-CoV-2 vaccination status were analyzed using descriptive statistics and time series analysis.

## Results

RSV was the most prevalent pathogen (5.5%), followed by SARS-CoV-2 (5.1%), Influenza A (4.4%), and Influenza B (1.7%). Individuals aged 45 and older were more likely to test positive for SARS-CoV-2. There was a significant difference in pathogen presence by occupation (p = 0.03), with health workers showing the highest prevalence of SARS-CoV-2 infection, while prevalence did not differ significantly by sex. Seasonal trends showed bimodal peaks for SARS-CoV-2, influenza A, and RSV, with the highest frequency observed between April 2023 to June 2023 and November 2023 to February 2024. However, Influenza B exhibited a single prolonged peak. SARS-CoV-2 vaccination was associated with a higher prevalence of SARS-CoV-2 (8.7% vs. 2.8%, p = 0.023) but not with influenza or RSV prevalence.

## Conclusion

The co-circulation of SARS-CoV-2, influenza, and RSV highlights Uganda's ongoing respiratory virus burden. Seasonal patterns and recurrent outbreaks underscore the need for sustained surveillance, targeted vaccination, and public health interventions to mitigate the impact of these pathogens, particularly in vulnerable populations.

## Background

Acute respiratory infections (ARIs) remain a leading cause of morbidity and mortality worldwide, particularly in low- and middle-income countries (LMICs) [1]. When these infections involve the lower respiratory tract referred to as lower respiratory tract infections (LRTIs), they can result in life-threatening complications, especially among the children, elderly, immunocompromised individuals, and those with underlying health conditions. The clinical presentation of ARIs ranges from mild upper respiratory symptoms such as cough, nasal congestion, and sore throat, to more severe manifestations including dyspnea, high fever, and respiratory distress. In LMICs, mortality associated with ARI-related LRTIs is disproportionately high due to constrained health system capacity, limited access to diagnostics, and delayed treatment [2–5]. Alongside influenza and SARS-CoV-2, RSV-associated infection also contributes significant burden, particularly affecting premature infants and the elderly, exacerbating global respiratory disease morbidity [6]. RSV exposure among these vulnerable populations often results in LRTI, hospitalizations, and adverse health outcomes [7]. Moreover, the co-circulation of SARS-CoV-2 with Influenza A and B viruses has been linked to exacerbated morbidity and mortality.

The global and Ugandan containment of the COVID-19 pandemic relied on a multifaceted strategy, encompassing the application of movement restrictions (lockdown) and the use of preventive vaccines [8,9]. However, despite the availability of effective

vaccines, influenza-associated burden due to endemic seasonal spread is increasingly recognized as a public health concern due to its seasonal recurrence [10] In Africa, mortality among hospitalized adults is approximately 12% [11]. Before the COVID-19 pandemic, viruses such as influenza and respiratory syncytial virus (RSV) were the predominant viral agents causing ARIs across many regions, including sub-Saharan Africa [12]. In Uganda and other East African countries, seasonal peaks of influenza and RSV have been observed during the rainy seasons, contributing to increased healthcare utilization during those months. However, the seasonal nature of these viruses is not consistent across the continent, as transmission patterns vary significantly by region, climate, and urbanization levels [3,4]. These recurrent viral epidemics strain under-resourced healthcare systems and affect socio-economic productivity, especially among working-age adults and caregivers.

The emergence of severe acute respiratory syndrome coronavirus 2 (SARS-CoV-2) in late 2019 significantly altered the global respiratory virus landscape. Although mitigation strategies such as lockdowns, social distancing, and SARS-Cov-2 vaccination campaigns were implemented to reduce transmission, the burden of COVID-19 persisted, especially in LMICs where vaccine coverage and access to intensive care services remain suboptimal [8,9]. In Uganda, like in many other LMICs, the health system struggled with diagnostic limitations, vaccine hesitancy, and logistical barriers during the pandemic response.

Recent evidence indicates that SARS-CoV-2 now co-circulates with other respiratory viruses such as influenza and RSV, especially during periods of increased viral transmission [13,14]. The simultaneous circulation of these viruses increases the complexity of clinical diagnosis, may exacerbate disease severity, and places additional pressure on healthcare systems. Co-infections have also been associated with longer hospital stays, higher rates of complications, and greater diagnostic uncertainty in settings with limited laboratory capacity. Despite this emerging pattern, there is a paucity of data on the co-circulation dynamics of SARS-CoV-2, influenza, and RSV in LMICs, including Uganda. Most existing studies were from high-income countries with robust surveillance and laboratory infrastructure, limiting the applicability of their findings to resource-limited settings. Moreover, the epidemiology of respiratory viruses in urban and peri-urban African contexts characterized by high population density, poor indoor air quality, limited access to health care, and a high burden of comorbidities such as HIV and tuberculosis remained poorly characterized [15,16].

Influenza viruses, once responsible for pandemics, now exhibit sporadic and epidemic transmission affecting tropical and temperate regions alike [17,18]. There is limited data on the epidemiologic dynamics of co-circulation of common respiratory viruses in the Ugandan population. In this study, we characterized the epidemiologic dynamics of SARS-CoV-2, influenza, and RSV co-circulation among participants with ARI in a post-COVID-19 pandemic setting in Uganda's central urban and peri-urban districts business districts from March 2023 to April 2024. These findings provide insights into viral transmission dynamics in a highly dynamic and populous community that could form public health policies, resource allocation, and effective strategies such as vaccine deployment in the tropical context. By identifying trends in age distribution, occupation-related exposure, and temporal variations in viral activity, the findings provide a crucial evidence base to inform future surveillance strategies, targeted vaccination campaigns, and clinical case management in similar LMIC settings.

## Methodology

### Study design and setting

This study employed an observational cross-sectional design conducted over a 13-month period, from March 2023 to April 2024. It was carried out at outpatient departments of selected public healthcare facilities across Kampala, Wakiso, and Mukono districts, areas that make up the Kampala Metropolitan Area in central Uganda. These urban and peri-urban facilities were selected based on high patient volume, wide geographical representation, and accessibility. The selected sites included Kawaala, Kisenyi, Kitebi, Kisugu, Kasangati, Kiswa, and Komamboga Health Centre IIIs, as well as Kiruddu National Referral Hospital and Mulago National Specialized Hospital. These facilities serve a diverse catchment

population estimated at over four million residents, encompassing individuals from low-income, densely populated neighborhoods, where respiratory infections are common due to overcrowding and environmental exposure.

## Study population and eligibility criteria

The study targeted adult patients aged 18 years and older who presented to participating outpatient departments with symptoms consistent with acute respiratory infection (ARI). The case definition for ARI was an acute onset (within the past 10 days) of respiratory symptoms including fever (≥38°C) and at least one of the following: cough, sore throat, or shortness of breath. Patients were enrolled consecutively after obtaining informed written consent.

Individuals were excluded from participation if they required hospitalization for severe illness, or if they presented with any signs of severe organ failure (including hepatic, renal, cardiac, or neurological impairment). Additionally, pregnant or lactating women were excluded. Patients meeting the criteria for severe acute respiratory illness (SARI) defined as ARI requiring inpatient care were also excluded from the study. The focus remained on ambulatory cases of mild to moderate ARI to capture community-level epidemiological trends of circulating respiratory pathogens.

## Specimen collection and referral

Nasopharyngeal (NP) and/or oropharyngeal (OP) swabs were obtained from each eligible participant by trained healthcare professionals following standard biosafety and specimen collection procedures. Each swab was immediately inserted into a tube containing viral transport medium (VTM), labeled with a unique identifier, and stored temporarily at 4–8°C before being transported to the Uganda Virus Research Institute (UVRI), specifically the National Influenza Centre (NIC), within 24 hours of collection. Proper cold-chain maintenance was observed during referral to preserve sample integrity.

## Laboratory testing and quality assurance

All samples were processed at the NIC using standardized molecular techniques. Total RNA was extracted from VTM samples using the Life River RNA extraction protocol with an elution volume of 50 µL. Real-time reverse transcription polymerase chain reaction (RT-PCR) was performed using the "CDC Flu SC2 multiplex assay", which simultaneously detects and differentiates SARS-CoV-2, Influenza A, and Influenza B. Testing was carried out on the ABI 7500 Real-Time PCR system, and each reaction included appropriate internal and external controls. RSV testing was conducted separately using the EXON RSV-specific RT-PCR assay on the same platform.

## Quality control

Each assay run included positive controls (known pathogen RNA) to confirm amplification integrity and negative controls (nuclease-free water) to rule out contamination. Human RNAseP was included as an internal control for sample adequacy. Results were considered valid only if control reactions met expected thresholds. Samples testing positive for any of the target viruses were recorded with corresponding cycle threshold (Ct) values.

## Data management and statistical analysis

Participant data, including demographic characteristics (age, sex, occupation), clinical symptoms, and laboratory findings, were collected and managed using the REDCap electronic data capture system hosted at Makerere University Lung Institute. Following data quality checks and de-identification, the dataset was exported to Stata version 15.0 (Stata Corp, College Station, TX, USA) for analysis.

Descriptive statistics were used to summarize the sociodemographic and clinical characteristics of the study population. Frequencies and proportions were computed for categorical variables, while means and standard deviations were used for continuous variables. Period prevalence for each pathogen (SARS-CoV-2, Influenza A, Influenza B, and RSV) was

calculated as the percentage of RT-PCR–positive cases among all tested individuals. To explore associations between viral positivity and key demographic or clinical factors, univariable analyses were conducted using chi-square or Fisher's exact tests as appropriate. Time series analysis was applied to detect seasonal patterns of pathogen circulation, using monthly stratification of cases to visualize epidemic curves. Results were presented with 95% confidence intervals and p-values, with statistical significance set at $p < 0.05$.

### Data analysis

All data were exported from REDCap into Stata version 15.0 (Stata Corp, College Station, TX, USA) for statistical analysis. Descriptive statistics were first performed to summarize demographic characteristics, clinical presentations, and laboratory-confirmed infections with respiratory viruses. Categorical variables were expressed as frequencies and percentages, while continuous variables (e.g., age and Ct values) were summarized using means and standard deviations.

For univariate analyses, we used Pearson's Chi-square test to assess associations between categorical variables (e.g., sex, age category, occupation) and RT-PCR-confirmed infection with SARS-CoV-2, Influenza A, Influenza B, and RSV. Where expected frequencies were <5 in any cell, Fisher's exact test was applied instead. Results were reported with p-values, and where applicable, 95% confidence intervals (CI) for proportions were calculated. The reference category for each variable was the most populous or clinically relevant group for example, the 18–34 age group was used as the reference for age-based comparisons due to its overrepresentation in the study sample. Age was categorized into three bands: 18–34, 35–44, and 45–83 years, based on observed age distribution and clinical relevance in respiratory disease susceptibility. While the 18–34 group was overrepresented (59.8%), the categories were preserved for internal consistency and comparability with national surveillance frameworks, though future studies should aim for a more evenly distributed sampling across age brackets.

A multivariable logistic regression analysis was subsequently conducted to control for potential confounding variables. Variables with a p-value <0.20 in univariate analyses were entered into the multivariate model to estimate adjusted odds ratios (aOR) and 95% confidence intervals for associations between socio-demographic factors and virus positivity. Model fitness was evaluated using the Hosmer-Lemeshow goodness-of-fit test.

### Ethical approval

Ethical approvals for this study were obtained from the Mulago National Referral Hospital Ethics Committee and the Uganda National Council of Science and Technology to conduct the study, with approval numbers MHREC 2344 and HS2548ES, respectively. The study was conducted in accordance with the Declaration of Helsinki

### Results

#### Participant characteristics

Out of the total 1,265 participants as shown in Table 1, the majority were aged 18–34 years (59.8%), with the remainder aged 35–44 years (19.8%) and 45–83 years (18.4%). Males comprised 62.1% of the sample. Table 1 presents the distribution of respiratory pathogens across age groups and sex. The overall prevalence estimates were: RSV 5.5% (95% CI: 4.3–6.9), SARS-CoV-2 5.1% (95% CI: 4.0–6.4), Influenza A 4.4% (95% CI: 3.3–5.6), and Influenza B 1.7% (95% CI: 1.1–2.6). A significant age-related difference was noted for SARS-CoV-2 positivity ($p = 0.007$), with individuals aged 45 years and above showing a higher prevalence (8.6%) compared to the reference 18–34 age group (4.9%). There were no statistically significant sex differences for any of the viral infections.

#### Seasonal trends

Epidemiological time-series analysis of the 13-month surveillance period revealed overlapping seasonal patterns of viral circulation. SARS-CoV-2 and Influenza A exhibited bimodal peaks, with peak activity in April-June and

**Table 1. Distribution of demographic characteristics and respiratory pathogen among screened participants.**

| Characteristics | Number of participants | Flu A | Flu B | SARS-CoV-2 | RSV |
|---|---|---|---|---|---|
| All (n)<br>Proportions %,(95% CI) | **1265** | **55**<br>4.4 (3.3, 5.6) | **22**<br>1.7 (1.1, 2.6) | **64**<br>5.1 (4.0, 6.4) | **69**<br>5.5 (4.3, 6.9) |
| Age category | | *P=0.156* | *P=0.202* | ***P=0.007*** | *0.953* |
| 18-34 | 756 (59.8%) | 31 (4.1%) | 16 (2.2%) | 37 (4.9%) | 43 (5.7%) |
| 35-44 | 250 (19.8%) | 17 (6.8%) | 4 (1.6%) | 5 (2.0%) | 14 (5.6%) |
| 45-83 | 233 (18.4%) | 7 (3.0%) | 1 (0.4%) | 20 (8.6%) | 11 (4.7%) |
| Missing | 26 (2.1%) | 0 (0.0%) | 1 (3.9%) | 2 (7.7%) | 1 (3.9%) |
| Sex | | *P=0.577* | *P=0.338* | *P=0.903* | *P=0.437* |
| Male | 785 (62.1%) | 23 (5.2%) | 11 (2.5%) | 21 (4.7%) | 26 (5.8%) |
| Female | 446 (35.3%) | 31 (4.0%) | 11 (1.4%) | 42 (5.4%) | 43 (5.5%) |
| Missing | 34 (2.7%) | 1 (2.9%) | 0 (0.0%) | 1 (2.9%) | 0 (0.0%) |

November-February. RSV also followed a bimodal pattern, while Influenza B showed a single prolonged peak during mid-year. The months of November and December had the lowest combined viral activity across all pathogens as shown in Fig 1.

Between March 2023 and April 2024, SARS-CoV-2, Influenza A, Influenza B, and RSV were co-circulating. Although Influenza B had the lowest prevalence and a single prolonged peak, the others exhibited two distinct peaks over a 12-month period. This bimodal pattern showed RSV peaking initially in April and again from June to October, SARS-CoV-2 in January and April, and Influenza A in April and September.

Summing up, overlapping waves of respiratory disease occurred in Jan (dominated by SARS-C0v-2), April-May (driven by Influenza A, SARS-CoV-2, and RSV), May (driven by SARS-CoV-2), and from July to October (driven by RSV), with November and December experiencing the lowest levels of activity. Overall, viral infection was most intense during the first half of the year, particularly from April to June, with notable co-circulation of Influenza A, SARS-CoV-2, and RSV. In contrast, viral infection across all four virus strains declined during the second half of the year

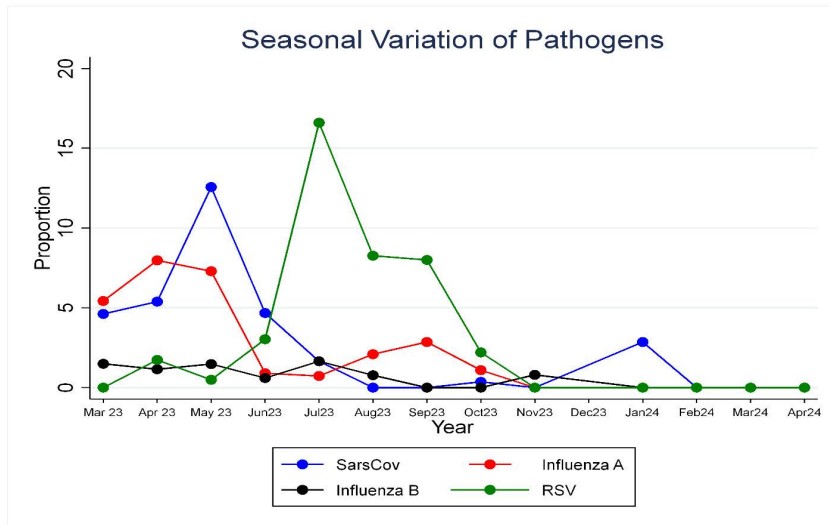

**Fig 1. Seasonal trend in prevalence of SARS-CoV-2, Influenza-A, Influenza-B, and RSV infections.**

## Clinical and occupational correlates

Out of 1,265 participants screened, In the 510 participants with complete clinical and occupational data. RSV remained the most frequently detected pathogen. Notably, health were workers demonstrated the highest prevalence of SARS-CoV-2 (23.8%, p = 0.03), significantly higher than other occupational categories such as business workers (6.5%) and students (5.3%), with unemployed/peasants used as the reference group. A statistically significant association was also observed between SARS-CoV-2 infection and the presence or absence of chills (p = 0.001), with infected individuals less likely to report chills (2.6% with chills vs. 11% without chills). Influenza A and B infections were more common among participants reporting sore throat (p = 0.002 and p = 0.037, respectively). No significant clinical symptom was strongly associated with RSV infection in univariate analysis as shown in Table 2.

Among participants with documented SARS-CoV-2 vaccination status, those who were vaccinated had a higher prevalence of SARS-CoV-2 infection (8.7%) compared to the unvaccinated group (2.8%, p = 0.023, Chi-square test). However, no significant associations were found between SARS-Cov-2 vaccination and the prevalence of Influenza A, B, or RSV. Further analysis confirmed that age ≥ 45 years (aOR: 2.31; 95% CI: 1.13–4.73) and being a healthcare worker (aOR: 3.85; 95% CI: 1.28–11.53) were independently associated with increased odds of SARS-CoV-2 infection. The association between SARS-Cov-2 vaccination and increased SARS-CoV-2 detection remained statistically significant (aOR: 2.94; 95% CI: 1.03–8.37), though this may reflect exposure bias rather than vaccine failure. No independent associations were found for Influenza or RSV after adjustment.

Except for Flu A, there were no differences in the prevalence of viral infection due to Flu B, SARS-CoV-2, and RSV based on occupational status. Apart from fever-related illness (febrile illness), which was common among participants, no distinct clinical symptoms were strongly linked to specific infections (etiologies such as SARS-CoV-2, Influenza A and B, or RSV). However, participants with SARS-CoV-2 were less likely to present with chills (2.6% of those with chills tested positive, compared to 11% without chills). Sore throat was uncommon among those with Influenza A or B, and no significant clinical associations were observed with RSV. Influenza A and B infections were more frequently detected among participants reporting sore throat compared with those without sore throat (p = 0.002 and p = 0.037, respectively). However, because sore throat was reported by a large proportion of the study population, it was not a distinguishing clinical feature of influenza infection, and most individuals with sore throat did not have laboratory-confirmed influenza A or B. Considering SARS-CoV-2 vaccination status, no significant differences were found in the prevalence of Flu A, Flu B, and RSV between the vaccinated and those who were not. However, SARS-CoV-2 infection was more common among those vaccinated against SARS-CoV-2 (8.7% versus 2.8%, P = 0.023).

## Discussion

This study investigated the circulation patterns of four major respiratory viruses; SARS-CoV-2, Influenza A, Influenza B, and respiratory syncytial virus (RSV) among adult outpatients with acute respiratory infection (ARI) in the Kampala Metropolitan Area between March 2023 and April 2024. Our findings contribute to the growing body of evidence documenting the ongoing burden and complexity of respiratory viruses in low- and middle-income countries (LMICs), particularly in post-pandemic urban African settings.

Among the viruses tested, RSV exhibited the highest individual prevalence (5.5%), surpassing that of SARS-CoV-2 (5.1%) and either Influenza A (4.4%) or B (1.7%) alone. However, when influenza A and B were combined, their total prevalence (6.1%) exceeded that of RSV. This distinction is important for contextualizing viral burden and underscores the continued relevance of influenza as a cause of ARI in adults. While RSV is classically associated with pediatric illness, our findings align with emerging evidence of RSV's substantial role in adult respiratory morbidity, particularly among the elderly and those with chronic comorbidities [6,7].

**Table 2. Distribution of baseline characteristics stratified by pathogen among study participants.**

| Characteristics | Number of participants, 510 | Flu A | Flu B | SARS-CoV-2 | RSV |
|---|---|---|---|---|---|
| Age category | | P=0.055 | P=0.867 | **P=0.003** | P=0.681 |
| 18-34 | 319 (62.5%) | 15 (4.7%) | 9 (2.8%) | 22 (6.9%) | 10 (3.1%) |
| 35-44 | 106 (20.8%) | 12 (11.3%) | 4 (3.8%) | 4 (3.8%) | 5 (4.7%) |
| 45-83 | 85 (16.7%) | 6 (7.1%) | 2 (2.4%) | 14 (16.5%) | 2 (2.4%) |
| Sex | | P=0.187 | **P=0.067** | P=0.423 | P=0.842 |
| Male | 223 (43.7%) | 18 (8.1%) | 10 (4.5%) | 15 (6.7%) | 7 (3.1%) |
| Female | 287 (56.3%) | 15 (5.2%) | 5 (1.7%) | 25 (8.7%) | 10 (3.5%) |
| Occupation | | **P=0.033** | P=0.459 | P=0.140 | P=0.838 |
| Unemployed/peasant | 71 (13.9%) | 2 (2.8%) | 0 (0.0%) | 6 (8.5%) | 2 (2.8%) |
| Business | 185 (36.3%) | 8 (4.3%) | 8 (4.3%) | 12 (6.5%) | 5 (2.7%) |
| Health worker | 21 (4.1%) | 0 (0.0%) | 0 (0.0%) | 5 (23.8%) | 0 (0.0%) |
| Student | 38 (7.5%) | 2 (5.3%) | 1 (2.6%) | 2 (5.3%) | 2 (5.3%) |
| Other | 195 (38.2%) | 21 (10.8%) | 6 (3.1%) | 15 (7.7%) | 8 (4.1%) |
| Health Facility | | P=0.131 | **P=0.029** | **P=0.003** | P=0.678 |
| Kiswa | 17 (3.3%) | 0 (0.0%) | 1 (5.9%) | 0 (0.0%) | 0 (0.0%) |
| Kitebi | 84 (16.5%) | 2 (2.4%) | 1 (1.2%) | 0 (0.0%) | 5 (6.0%) |
| Kirudu | 97 (19.0%) | 10 (10.3%) | 4 (4.1%) | 15 (15.5%) | 4 (4.1%) |
| Mulago | 43 (8.4%) | 2 (4.7%) | 2 (4.7%) | 6 (14.0%) | 1 (2.3%) |
| Kisenyi | 6 (1.2%) | 2 (33.3%) | 2 (33.3%) | 0 (0.0%) | 0 (0.0%) |
| Kawala | 127 (24.9%) | 9 (7.1%) | 1 (0.8%) | 9 (7.1%) | 5 (3.9%) |
| Kasangati | 18 (3.5%) | 1 (5.6%) | 0 (0.0%) | 0 (0.0%) | 0 (0.0%) |
| Kisugu | 18 (3.5%) | 0 (0.0%) | 0 (0.0%) | 2 (11.1%) | 1 (5.6%) |
| Nkomamboga | 100 (19.6%) | 7 (7.0%) | 4 (4.0%) | 8 (8.0%) | 1 (1.0%) |
| Clinical presentation | | | | | |
| Temperature | | | | | |
| Temp>37.8 | 510 (100.0%) | 33 (6.5%) | 15 (2.9%) | 40 (7.8%) | 17 (3.3%) |
| History of fever | | P=0.581 | P=0.715 | **P=0.020** | **P=0.020** |
| No | 193 (37.8%) | 11 (5.7%) | 5 (2.6%) | 22 (11.4%) | 11 (5.7%) |
| Yes | 317 (62.2%) | 22 (6.9%) | 10 (3.2%) | 18 (5.7%) | 6 (1.9%) |
| Cough | | P=0.417 | p>0.999 | **P=0.019** | P=0.761 |
| No | 8 (1.6%) | 1 (12.5%) | 0 (0.0%) | 3 (37.5%) | 0 (0.0%) |
| Yes | 502 (98.4%) | 32 (6.4%) | 15 (3.0%) | 37 (7.4%) | 17 (3.4%) |
| Sore throat | | **P=0.002** | **P=0.037** | P=0.069 | P=0.599 |
| No | 68 (13.3%) | 11 (16.2%) | 5 (7.4%) | 9 (13.2%) | 2 (2.9%) |
| Yes | 442 (86.7%) | 22 (5.0%) | 10 (2.3%) | 31 (7.0%) | 15 (3.4%) |
| Headache | | P=0.782 | P=0.702 | P=0.368 | P=0.049 |
| No | 159 (31.2%) | 11 (6.9%) | 4 (2.5%) | 15 (9.4%) | 9 (5.7%) |
| Yes | 351 (68.8%) | 22 (6.3%) | 11 (3.1%) | 25 (7.1%) | 8 (2.3%) |
| Chills | | P=0.613 | P=0.738 | **P=0.001** | P=0.405 |
| No | 319 (62.5%) | 22 (6.9%) | 10 (3.1%) | 35 (11.0%) | 9 (2.8%) |
| Yes | 191 (37.5%) | 11 (5.8%) | 5 (2.6%) | 5 (2.6%) | 8 (4.2%) |
| Malaise | | P=0.461 | P=0.156 | P=0.882 | P=0.392 |
| No | 248 (48.6%) | 14 (5.6%) | 10 (4.0%) | 19 (7.7%) | 10 (4.0%) |
| Yes | 262 (51.4%) | 19 (7.3%) | 5 (1.9%) | 21 (8.0%) | 7 (2.7%) |
| VL/CT value, mean (SD) | | 28.9 (4.5) | 30.4 (4.5) | 29.4 (6.5) | 26.6 (5.8) |
| SARS-CoV-2 Vaccination Status | | P=0.665 | P=0.485 | **P=0.023** | P=0.618 |

*(Continued)*

**Table 2.** (Continued)

| Characteristics | Number of participants, 510 | Flu A | Flu B | SARS-CoV-2 | RSV |
|---|---|---|---|---|---|
| No | 109 (24.0%) | 6 (5.5%) | 2 (1.8%) | 3 (2.8%) | 3 (2.8%) |
| Yes | 345 (76.0%) | 23 (6.7%) | 9 (2.6%) | 30 (8.7%) | 10 (2.9%) |

The higher RSV prevalence in adults compared to individual influenza subtypes may reflect post-pandemic shifts in viral ecology or immunity gaps caused by reduced RSV exposure during COVID-19 lockdowns [13]. Similar RSV trends in adults have been observed in Europe and North America during the post-COVID period, and these shifts warrant further surveillance in LMICs [7]. Nonetheless, our results also suggest that influenza viruses remain prominent contributors to outpatient ARIs and should not be deprioritized in surveillance and vaccine planning.

Notably, SARS-CoV-2 infection was significantly more prevalent among participants aged ≥45 years and among healthcare workers, consistent with previous reports highlighting increased risk due to occupational exposure and age-related vulnerability [19,20]. Surprisingly, participants who were vaccinated against SARS-CoV-2 had a higher infection prevalence than unvaccinated individuals. This counterintuitive finding may reflect increased exposure risk among vaccinated populations such as healthcare personnel behavioral risk compensation, rather than vaccine failure [9]. It may also reflect waning immunity or mismatch between circulating variants and vaccine-induced protection [21].

Temporal analysis showed distinct periods of increased virus circulation: – notably between April – June and November – February for SARS-CoV-2, Influenza A, and RSV. These periods of heightened activity, while suggestive of recurring peaks, should not be labeled as "seasonal trends", as this study spanned only one year and lacked inter-annual comparison. Similar bimodal virus peaks have been observed in tropical regions, influenced by climatic conditions such as rainfall and humidity [22,23]. The alignment of virus surges during the wet months in Uganda underscores the need to align vaccine campaigns and health preparedness efforts with these locally relevant periods of increased respiratory pathogen activity. These findings suggest that vaccination strategies in Uganda should move beyond a single-pathogen focus and be informed by periods of heightened respiratory virus circulation. The observed burden of influenza and SARS-CoV-2 among adults particularly older individuals and healthcare workers supports prioritizing these groups for timely vaccination ahead of anticipated peaks in virus activity. Strengthening adult influenza vaccination and maintaining risk-based SARS-CoV-2 booster strategies may enhance protection in urban settings where multiple respiratory viruses co-circulate.

Importantly, this study did not assess transmission dynamics, such as reproductive numbers or contact-based risk modeling. Rather, it focused on describing circulation patterns and epidemiologic associations, based on RT-PCR-confirmed pathogen detection. While valuable for public health planning, further studies are needed to quantify person-to-person transmission and virus interaction within communities.

This study had several limitations. The most significant, the participants were drawn from an existing clinical trial, which likely introduced selection bias. As trial participants were subject to specific inclusion and exclusion criteria, they may not be fully representative of the general adult outpatient population in Kampala or Uganda. For example, severely ill patients and children were excluded, limiting generalizability across all age groups and disease severities. Second, due to missing data on clinical and occupational characteristics, the analytical sample for some variables was reduced from 1,265–510, which may have affected the power and reliability of subgroup analyses. Third, the reliance on self-reported data for symptoms and SARS-Cov-2 vaccination status may have introduced recall bias or misclassification.

Despite these limitations, the study offers important insights into the co-circulation of SARS-CoV-2, influenza viruses, and RSV among urban adult outpatients. It underscores the need for continuous, year-round virologic surveillance in LMIC settings and provides a basis for designing more comprehensive, population-level studies that include children, hospitalized patients, and rural populations.

This study did not assess transmission dynamics, such as reproductive numbers or contact-based risk modeling, and instead focused on describing circulation patterns and epidemiologic associations based on RT-PCR–confirmed pathogen detection. Participants were recruited from an existing clinical research platform, which may have introduced selection bias and limited representativeness of the broader adult outpatient population in study area. The exclusion of children and severely ill patients further restricts generalizability across age groups and disease severity spectra. Additionally, missing clinical and occupational data reduced the analytical sample for some variables, potentially affecting statistical power and precision of subgroup analyses, while reliance on self-reported symptoms and SARS-CoV-2 vaccination status may have introduced recall bias or misclassification. The 13-month observation period also limits inference regarding long-term or inter-annual seasonal patterns.

Despite these limitations, the study provides important insight into the co-circulation of SARS-CoV-2, influenza viruses, and RSV among adult outpatients in an urban LMIC setting. The findings underscore the need for sustained, year-round virologic surveillance and support the development of broader population-based studies incorporating pediatric populations, hospitalized patients, rural communities, and transmission modeling approaches to better inform public health planning.

## Conclusions

This study demonstrates concurrent circulation of SARS-CoV-2, influenza A/B, and RSV among adults with acute respiratory infection in outpatient settings in the Kampala Metropolitan Area. RSV was the most frequently detected single pathogen, while the combined burden of influenza A and B exceeded that of RSV. Older adults (≥45 years) and healthcare workers had a higher likelihood of SARS-CoV-2 infection. Distinct temporal peaks in virus detection were observed during the study period, indicating periods of increased respiratory virus activity in this urban LMIC setting.

## Acknowledgments

We extend our heartfelt appreciation to the Makerere University Lung Institute CONAT Implementation Team for their invaluable contributions to this study. Special thanks go to the dedicated Medical Officers Wali Najja Daud, Joseph Byamugisha, Nelson Twinamatsiko, and Hope Atuhaire for their clinical expertise and oversight. We were equally grateful to the Study Nurses; Nassolo Hellen Martha, Sarah Apolot, Lillian Namyenya, and Sarah Namususwa whose commitment and professionalism greatly supported the study's success. We acknowledge the diligent efforts of Laboratory Technicians Vincent Wadda and Ignitius Okello, as well as Acan Winnie, our Laboratory Runner, for ensuring timely and accurate sample handling. We further appreciate the tireless support of our Research Facilitators; Nakabuye Charity, Kinene Enock, Wakoko Anold, Sendi Gerald, Okello Francis Moses, Mufumba Siraje, Opollot Deogracious, Kaggwa Andrew, and Oditai Paul for their critical role in data collection and coordination across study sites. Most importantly, we express our sincere gratitude to all the study participants, without whom this research would not have been possible. We also thank the management of Mulago National Referral Hospital and the Kampala Capital City Authority (KCCA) Public Health Administration, and Wakiso District Health Administration for granting us access to the respective health facilities, thereby enabling the successful implementation of this study.

**Consent for publication:** The authors consented to the publication of the content of the material here.

## Author contributions

**Conceptualization:** Barnabas Bakamutumaho, Noah Kiwanuka, Moses Joloba, Moses Ocan, Eddie Wampande, Jane Nakibuuka, John Lusiba, Joseph Okia, Pauline Byakika-Kibwika, Bruce Kirenga.

**Data curation:** Noah Kiwanuka.

**Formal analysis:** Levicatus Mugenyi.

**Investigation:** Barnabas Bakamutumaho, Jacqueline Kyosiimire Lugemwa, Bruce Kirenga.

**Methodology:** Barnabas Bakamutumaho, Bruce Kirenga.

**Supervision:** Bruce Kirenga.

**Writing – original draft:** James Arinaitwe.

**Writing – review & editing:** Barnabas Bakamutumaho, Noah Kiwanuka, Patricia Alupo, Aida N Kawuma, Karen Ndahura, Maria Sekimpi, Eva Akurut, Winters Muttamba, Darius Owachi, Pauline Byakika-Kibwika, Bruce Kirenga.

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
