## [Decision Letter · Decision Letter 0]

16 Dec 2025

PONE-D-25-42419The epidemiology of SARS-CoV-2, Influenza, and Respiratory Syncytial Virus circulation among adult patients with acute respiratory infection in Kampala Metropolitan Area in Uganda.PLOS One

Dear Dr. Kirenga

Thank you for submitting your manuscript to PLOS ONE. After careful consideration, we feel that it has merit but does not fully meet PLOS ONE’s publication criteria as it currently stands. Therefore, we invite you to submit a revised version of the manuscript that addresses the points raised during the review process. Please submit your revised manuscript by Jan 30 2026 11:59PM. If you will need more time than this to complete your revisions, please reply to this message or contact the journal office at plosone@plos.org. Please include the following items when submitting your revised manuscript:

We look forward to receiving your revised manuscript.

Kind regards,

Haitham Mohamed Amer, PhD

Academic Editor

PLOS One

Journal Requirements:

1. When submitting your revision, we need you to address these additional requirements. Please ensure that your manuscript meets PLOS ONE's style requirements, including those for file naming. The PLOS ONE style templates can be found at https://journals.plos.org/plosone/s/file?id=wjVg/PLOSOne_formatting_sample_main_body.pdf and https://journals.plos.org/plosone/s/file?id=ba62/PLOSOne_formatting_sample_title_authors_affiliations.pdf 2. Thank you for stating the following financial disclosure: The study was funded by the Government of Uganda through the secretariat of Science, Innovation and Technology - office of the president   Please state what role the funders took in the study.  If the funders had no role, please state: "The funders had no role in study design, data collection and analysis, decision to publish, or preparation of the manuscript." If this statement is not correct you must amend it as needed. Please include this amended Role of Funder statement in your cover letter; we will change the online submission form on your behalf. 3. In the online submission form, you indicated that “Any material relevant to the publication is available upon reasonable request to the corresponding author.” All PLOS journals now require all data underlying the findings described in their manuscript to be freely available to other researchers, either a. In a public repository, b. Within the manuscript itself, or c. Uploaded as supplementary information.This policy applies to all data except where public deposition would breach compliance with the protocol approved by your research ethics board. If your data cannot be made publicly available for ethical or legal reasons (e.g., public availability would compromise patient privacy), please explain your reasons on resubmission and your exemption request will be escalated for approval. 4. When completing the data availability statement of the submission form, you indicated that you will make your data available on acceptance. We strongly recommend all authors decide on a data sharing plan before acceptance, as the process can be lengthy and hold up publication timelines. Please note that, though access restrictions are acceptable now, your entire data will need to be made freely accessible if your manuscript is accepted for publication. This policy applies to all data except where public deposition would breach compliance with the protocol approved by your research ethics board. If you are unable to adhere to our open data policy, please kindly revise your statement to explain your reasoning and we will seek the editor's input on an exemption. Please be assured that, once you have provided your new statement, the assessment of your exemption will not hold up the peer review process. 5. Your ethics statement should only appear in the Methods section of your manuscript. If your ethics statement is written in any section besides the Methods, please delete it from any other section. 6. Please include your tables as part of your main manuscript and remove the individual files. Please note that supplementary tables (should remain/ be uploaded) as separate "supporting information" files. 7. If the reviewer comments include a recommendation to cite specific previously published works, please review and evaluate these publications to determine whether they are relevant and should be cited. There is no requirement to cite these works unless the editor has indicated otherwise.

Reviewers' comments:

Reviewer's Responses to Questions

Comments to the Author

1. Is the manuscript technically sound, and do the data support the conclusions?

Reviewer #1: Yes

Reviewer #2: Yes

2. Has the statistical analysis been performed appropriately and rigorously? 

Reviewer #1: Yes

Reviewer #2: Yes

3. Have the authors made all data underlying the findings in their manuscript fully available?

Reviewer #1: Yes

Reviewer #2: Yes

4. Is the manuscript presented in an intelligible fashion and written in standard English?

Reviewer #1: Yes

Reviewer #2: Yes

5. Review Comments to the Author

Reviewer #1: This is a very nice review of respiratory virus surveillance in a large urban area in Uganda highlighting the co-circulation dynamics of SARS-CoV-2, influenza, and RSV. The paper is well written and clear. I think it would be very suitable for publication, once certain questions about data comprehensive, statistical bias and exclusion criteria are addressed.

Some minor comments:

Lines 115-116: I would not say that influenza viruses are no longer responsible for pandemics, which they clearly could be again

Lines 150-152: Why did you exclude patients requiring admission, as those with severe acute respiratory illness requiring inpatient care would still give you information on community circulation, and excluding them could bias prevalence depending on differential severity. Why also were pregnant / lactating women excluded - as this could bias certain professions and would be a very important group to characterize. I assume due to the trial they were drawn from, but I would make this clear and highlight the specific limitations here in understanding true prevalence.

Lines 189-191: Here you say only univariable analysis was performed, however you did continue to multivariable analysis.

Lines 246-248: It could be that healthcare workers were more easily accessing clinics due to place of work that may account for the high prevalence of SARS-CoV-2 as well as occupational exposure this should be accounted for in discussion

Line 259: Was this vaccination for SARS-Cov_2, or for influenza, I assume none for RSV

Line 265: This belongs in discussion - "though this may reflect exposure bias rather than vaccine failure"

Line 274: Use FLU A but Flu B - should have consistent throughout article

Line 280: You say that 'Sore throat was uncommon among those with Influenza A or B' but this goes against previous statement that 'Influenza A and B infections were more common among 253 participants reporting sore throat (p = 0.002 and p = 0.037, respectively).'

Line 320: SARS-CoV-2 could also have been detected as a bystander in vaccinated symptomatic individuals who may have had another unidentified infection alongside a causative agent

Lines 347-350: Could this drop-off in reporting of certain variables lead to differential bias, clearly possible. Was there uniformity in lack of reporting by age or sex?

Line 357+: I would like to see what implications you think this might have for vaccination strategies in Uganda

Line 372: prepweredness is spelt incorrectly

Table 1: Can you also comment on any coinfections observed, and include this within the text

Table 2: Can you add totals out of 510 so it is easy to see how many missing variables there are? Or did you only include those 510 that detailed all the questions, if so as above a view by age/sex of who did not answer would be useful.

Reviewer #2: Review (PLOS One)

The authors said that “acute respiratory infections (ARI) caused by viruses such as SARS-CoV-2, influenza, and respiratory syncytial virus (RSV) posed significant public health challenges, particularly in low-income countries. Understanding the co-circulation dynamics of these pathogens is crucial for effective public health monitoring, surveillance, and control interventions”. The authors have raised an important issue with the aim of characterizing the epidemiologic dynamics of SARS-CoV-2, influenza, and RSV co-circulation among participants with ARI in a post-COVID-19 pandemic setting in Uganda's central urban and peri-urban districts business districts.

However, there are some concerns:

Methodology

Based on the methodology used, I do not understand why the authors qualified this study as a retrospective cohort study. This should clearly be explained.

Line 228-232: The clarifications to justify Table 2 and Table 1 should not be done in this section since they have been repeated in the result section.

Results

Line 244 -260 : these two paragraphs should be combined since the Multivariate analysis is only a technical tool empowering the observed results. Additionally in this section all sentences related to explain some results are note expected (e.g. line 266).

Lines 274-285: this paragraph does not fit with the “Seasonal trends” paragraph. May be the authors should reorganize all these information’s in the “Clinical and occupational correlates” by avoiding repetition of the same information’s (e.g. Lines 246,247, 248-274, 275).

Conclusion

The conclusion should be shortly formulated and possible limitations of the study should be stated in a special paragraph.

6. PLOS authors have the option to publish the peer review history of their article (what does this mean?). If published, this will include your full peer review and any attached files.

Do you want your identity to be public for this peer review? For information about this choice, including consent withdrawal, please see our Privacy Policy.

Reviewer #1:  Yes: Colin Brown

Reviewer #2: No

---

## [Author Response · Author response to Decision Letter 1]

11 Jan 2026

All comments have been addressed as per "Reviewed manuscript" with track changes and clean copy attached. "Point by Point" reply has been attached among the documents

The study was funded by the Government of Uganda through the Secretariat of Science, Technology and Innovation, Office of the President. The funders had no role in the study design; data collection, analysis, or interpretation; decision to publish; or preparation of the manuscript.

---

## [Decision Letter · Decision Letter 1]

16 Feb 2026

PONE-D-25-42419R1The epidemiology of SARS-CoV-2, Influenza, and Respiratory Syncytial Virus circulation among adult patients with acute respiratory infection in Kampala Metropolitan Area in Uganda.PLOS One

Dear Dr. Kirenga,

Thank you for submitting your manuscript to PLOS ONE. After careful consideration, we feel that it has merit but does not fully meet PLOS ONE’s publication criteria as it currently stands. Therefore, we invite you to submit a revised version of the manuscript that addresses the points raised during the review process.

We look forward to receiving your revised manuscript.

Kind regards,

Haitham Mohamed Amer, PhD

Academic Editor

PLOS One

Journal Requirements:

Reviewers' comments:

Reviewer's Responses to Questions

Comments to the Author

1. If the authors have adequately addressed your comments raised in a previous round of review and you feel that this manuscript is now acceptable for publication, you may indicate that here to bypass the “Comments to the Author” section, enter your conflict of interest statement in the “Confidential to Editor” section, and submit your "Accept" recommendation.

Reviewer #1: All comments have been addressed

Reviewer #2: All comments have been addressed

2. Is the manuscript technically sound, and do the data support the conclusions?

Reviewer #1: Yes

Reviewer #2: Yes

3. Has the statistical analysis been performed appropriately and rigorously? 

Reviewer #1: Yes

Reviewer #2: Yes

4. Have the authors made all data underlying the findings in their manuscript fully available?

Reviewer #1: Yes

Reviewer #2: Yes

5. Is the manuscript presented in an intelligible fashion and written in standard English?

Reviewer #1: Yes

Reviewer #2: Yes

6. Review Comments to the Author

Reviewer #1: Thank you for addressing my previous comments and that of the other reviewer, which have strengthened it considerably.

The addition of a limitations section is welcome, but it does not sit after the conclusions and should rather be integrated into the discussion section.

Reviewer #2: The authors have raised an important issue with the aim of characterizing the epidemiologic dynamics of SARS-CoV-2, influenza, and RSV co-circulation among participants with ARI in a post-COVID-19 pandemic setting in Uganda's central urban and peri-urban districts business districts.

All the comments/edits raised have been attended. I have no further comments

7. PLOS authors have the option to publish the peer review history of their article (what does this mean?). If published, this will include your full peer review and any attached files.

Do you want your identity to be public for this peer review? For information about this choice, including consent withdrawal, please see our Privacy Policy.

Reviewer #1:  Yes: Colin Brown

Reviewer #2: No

---

## [Author Response · Author response to Decision Letter 2]

17 Feb 2026

Reviewer #1 Comment:

“Thank you for addressing my previous comments and that of the other reviewer, which have strengthened it considerably. The addition of a limitations section is welcome, but it does not sit after the conclusions and should rather be integrated into the discussion section.”

Response:

We thank the reviewer for this important observation. In response, we have revised the manuscript by removing the standalone “Study limitations” section that appeared after the Conclusions. The limitations have now been fully integrated into the final section of the Discussion, immediately before the concluding transition paragraph specifically, from Line 353- 368; We believe this revision strengthens the manuscript’s interpretive clarity and appropriately addresses the reviewer’s concern. We once again thank the Editor and Reviewer for their thoughtful feedback and for the opportunity to improve our work further. We hope the revised manuscript meets the journal’s expectations and are happy to provide any additional clarifications if needed.

---

## [Editor Report · Decision Letter 2]

27 Feb 2026

The epidemiology of SARS-CoV-2, Influenza, and Respiratory Syncytial Virus circulation among adult patients with acute respiratory infection in Kampala Metropolitan Area in Uganda.

PONE-D-25-42419R2

Dear Dr. Kirenga,

We’re pleased to inform you that your manuscript has been judged scientifically suitable for publication and will be formally accepted for publication once it meets all outstanding technical requirements.

Kind regards,

Haitham Mohamed Amer, PhD

Academic Editor

PLOS One

---

## [Editor Report · Acceptance letter]

PONE-D-25-42419R2

PLOS One

Dear Dr. Kirenga,

I'm pleased to inform you that your manuscript has been deemed suitable for publication in PLOS One. Congratulations! Your manuscript is now being handed over to our production team.

Kind regards,

on behalf of

Dr. Haitham Mohamed Amer

Academic Editor

PLOS One